# Unlocking the Resistance to Anti-HER2 Treatments in Breast Cancer: The Issue of HER2 Spatial Distribution

**DOI:** 10.3390/cancers15051385

**Published:** 2023-02-22

**Authors:** Federica Giugliano, Ambra Carnevale Schianca, Chiara Corti, Mariia Ivanova, Nadia Bianco, Silvia Dellapasqua, Carmen Criscitiello, Nicola Fusco, Giuseppe Curigliano, Elisabetta Munzone

**Affiliations:** 1Division of New Drugs and Early Drug Development for Innovative Therapies, European Institute of Oncology, IRCCS, 20141 Milan, Italy; 2Department of Oncology and Hemato-Oncology, University of Milan, 20122 Milan, Italy; 3Division of Medical Senology, IEO European Institute of Oncology, IRCCS, 20141 Milan, Italy; 4Division of Pathology, IEO European Institute of Oncology, IRCCS, 20141 Milan, Italy

**Keywords:** HER2, breast cancer, spatial distribution, heterogeneity, resistance

## Abstract

**Simple Summary:**

Breast cancer is the most common cancer in women. Approximately 15% of breast cancers harbour an amplification of the *ERBB2* gene and/or an overexpression of the HER2 protein and are thus classified as HER2-positive. However, HER2 protein expression could be heterogeneous, showing different patterns of spatial distribution. This feature, also called “spatial heterogeneity” may potentially affect treatment, response, and assessment of HER2 status, ultimately impacting the best treatment strategy. The activity of some new pharmacological agents, belonging to the group of antibody–drug conjugates, may represent an opportunity for overcoming this issue. In this review, we summarize the available evidence on HER2 heterogeneity and spatial distribution and how they may affect current available treatment choices.

**Abstract:**

Approximately 15% of breast cancers are classified as HER2-positive, with an amplification of the *ERBB2* gene and/or an overexpression of the HER2 protein. Up to 30% of HER2-positive breast cancers shows heterogeneity in HER2 expression and different patterns of spatial distribution, i.e., the variability in the distribution and expression of the HER2 protein within a single tumour. Spatial heterogeneity may potentially affect treatment, response, assessment of HER2 status and consequently, may impact on the best treatment strategy. Understanding this feature can help clinicians to predict response to HER2-targeted therapies and patient outcomes, and to fine tune treatment decisions. This review summarizes the available evidence on HER2 heterogeneity and spatial distribution and how this may affect current available treatment choices, exploring possible opportunities for overcoming this issue, such as novel pharmacological agents, belonging to the group of antibody–drug conjugates.

## 1. Introduction

Approximately 15% of breast cancers (BCs) harbour an amplification of the *ERBB2* gene, which encodes human epidermal growth factor receptor 2 (HER2) [1]. Four main biological entities of BC could be identified, based on the gene expression profile: luminal A, luminal B, HER2-enriched and basal-like [2]. In clinical practice, a surrogate classification is adopted, based on the evaluation of three key predictive biomarkers: two hormone receptors (HR), namely estrogen receptor (ER) and progesterone receptor (PgR), and HER2 overexpression and/or amplification [3]. Thus, experts commonly define three different BC subtypes: luminal-like (ER-positive and/or PgR-positive and HER2-negative), HER2-positive (HER2 overexpression/amplification, with any ER and PgR expression), and triple negative (ER-negative, PgR-negative and HER2-negative) tumours. According to the latest ASCO/CAP guidelines (2018), the evaluation of HER2 status is defined by an immunohistochemistry (IHC) score as follows: 0 or 1+ for HER2-negative, 2+ for HER2-equivocal and 3+ for HER-positive [4]. HER2-equivocals are further examined by in situ hybridization (ISH) assay to categorize tumours as either HER2-negative or HER2-positive [4,5] (Table 1).

ASCO/CAP guidelines state the predictive role of HER2, helping clinicians to identify patients with HER2-positive tumours who would benefit from anti-HER2 targeted treatment, compared to patients affected by HER2-negative BC [6,7]. Indeed, targeted therapies capable of binding HER2 and blocking the downstream signalling, such as trastuzumab, pertuzumab, lapatinib and trastuzumab emtansine (T-DM1), have significantly changed the prognosis of HER2-positive BC [3,6,8,9,10,11]. The mechanisms of resistance and sensitivity to anti-HER2 therapies in HER2-positive BC have been thoroughly investigated and reported [12,13,14], however, in this context little is known about the role of HER2 heterogeneity defined by ISH [15,16]. Indeed, HER2 expression could be heterogeneous, showing different patterns of spatial and temporal distribution. HER2 temporal heterogeneity is beyond the scope of this review [4,17]. 

HER2 spatial heterogeneity refers to the variability in the distribution and expression of the HER2 protein within a single tumour. This means that different areas within the same tumour may have different levels of HER2 expression, thus making the targeting of the protein potentially less effective. This is one of the reasons that may lead to treatment resistance. Spatial heterogeneity may also make it more difficult to accurately assess the HER2 status of a tumour and, accordingly, to determine the best treatment approach.

Understanding the spatial heterogeneity of HER2 expression within a tumour can be important to predict response to HER2-targeted therapies, to fine tune treatment decisions and to predict patient outcomes [17,18]. Moreover, the different patterns of HER2 spatial distribution could play a role in the efficacy of anti-HER2 drugs. The activity of some new pharmacological agents, belonging to the group of antibody–drug conjugates (ADC), may represent an opportunity for overcoming this issue. This review summarizes the available evidence on HER2 heterogeneity and spatial distribution and how they may affect current available treatment choices. 

## 2. Overview of the Current Management of HER2-Positive Breast Cancer 

The current management of HER2-positive BC in early and in metastatic settings relies on anti-HER2 targeted therapies. In the early setting, the choice of (neo)adjuvant treatment is based upon the expected tumour sensitivity to pre-specified agents, individual risk of relapse and long-term toxicities. For tumours with a diameter less than 1 cm and without clinical lymph nodes involvement, surgery can be proposed. If a pathological stage I is confirmed, a treatment de-escalation could be planned (weekly paclitaxel for 12 administrations plus trastuzumab for 1 year), according to the results of the APT trial (3-yr invasive disease free survival (iDFS) 98.7%, confidence interval (CI) = 97.6–99.8) [19]. In case of an (unexpected) pathological stage II or III, the preferred choice is adjuvant trastuzumab (plus pertuzumab) for 1 year with docetaxel/carboplatin (TC) or anthracyclines/cyclophosphamide/paclitaxel (ACT) [20]. For tumours of at least 1–2 cm in greatest diameter, a neoadjuvant approach is the standard of care with chemotherapy plus trastuzumab +/− pertuzumab, with the possibility of fine tuning the treatment choice based on the individual risk of relapse [20]. If pathological complete response (pCR) is not achieved after neoadjuvant treatment, T-DM1 has to be offered based on the results of the KATHERINE trial (3-yr iDFS 88.3% vs. 77.0%, hazard ratio 0.50 CI 0.39–0.64, *p* < 0.001) [21]. In case a pCR is obtained, trastuzumab (plus pertuzumab) should be administered for a total of 12 months. For triple positive tumours (i.e., HR-positive and HER2-positive tumours), endocrine therapy should be added as per international guidelines [20].

In the advanced setting, the CLEOPATRA clinical trial sets a new standard of care, represented by pertuzumab, trastuzumab and docetaxel, followed by maintenance pertuzumab/trastuzumab (median overall survival (mOS) 56.5 months vs. 40.8 months, hazard ratio 0.68 CI 0.56–0.84, *p* > 0.001) [3,22,23]. Upon disease progression, patients could receive a new treatment option with trastuzumab deruxtecan (T-DXd) or tucatinib/capecitabine/trastuzumab (preferred choice in case of active brain metastases) [3]. Indeed, data from the DESTINY-Breast03 trial established T-DXd as the second-line preferred choice in patients previously treated with a taxane and trastuzumab, given the benefit in both progression-free survival [PFS] and OS: for PFS, 28.8 months vs. 6.8 months compared to T-DM1 (hazard ratio 0.33, 95% CI 0.26–0.43, *p* < 0.0001); OS not reached (CI 40.5 months–not estimable) vs. OS not reached (34.0 months—not estimable), hazard ratio 0.64, (CI 0.47–0.87, *p* = 0.0037) [24,25]. Even if the randomized phase II HER2CLIMB enrolled patients who had previously received trastuzumab and T-DM1, the PFS and OS benefits in patients with active or stable brain metastasis suggest its possible use for selected patients in this setting [26]. At the time being, prospective data on the best treatment sequence for patients in third line and beyond are lacking. Certainly, there is the possibility to use tucatinib or T-DXd if not previously used [3]. The use of T-DM1 is possible as well [3]. Later lines may include the tyrosine kinase inhibitor (TKI) lapatinib, preferably in combination with capecitabine, and of the pan-HER TKI neratinib plus chemotherapy (Food and Drug Administration [FDA] approved, not European Medical Agency (EMA) approved), margetuximab might be used as well (FDA approved, not EMA approved) [3,27]. Another, although old-fashion possibility, is to keep using trastuzumab beyond disease progression, but in combination with a different chemotherapeutic agent (e.g., vinorelbine, capecitabine, eribulin) [3]. Endocrine therapy could be added to anti-HER2 treatments if BC also expresses HR [3]. 

## 3. The Resistance to Anti-HER2 Treatments

HER2 is a transmembrane protein receptor belonging to the HER family [8,28]. All members of this family are receptor tyrosine kinases (RTK) with an analogous structure, characterized by an extracellular ligand-binding domain, a transmembrane domain, and an intracellular tyrosine kinase domain. The binding between HER receptors and their ligands induces both homodimeric and heterodimeric interactions between family members, with consequent activation of downstream pathways involving phosphatidylinositol 3-kinase (PI3K) and RAS. When PI3K is stimulated, it leads to activation of protein kinase B (AKT) and mammalian target of rapamycin (mTOR), promoting cell proliferation, growth, and survival [28,29]. RAS activates downstream proteins, namely the RAF, MEK and ERK. The oncogenic role of HER2 is mainly associated with the *ERBB2* gene overexpression (located on chromosome 17q21), increasing the number of HER2 heterodimers on the cell’s surface with the hyperactivation of oncogenic pathways, ultimately driving cell cycle progression, angiogenesis, invasiveness and metabolic reprogramming [29,30]. In this section, the main mechanisms of resistance to anti-HER2 targeted therapy are summarized [12,31] (Figure 1).

### 3.1. Impaired Binding to HER2

To function properly, each drug should be capable of accessing its binding site located on the HER2 protein. Low levels of HER2 and HER2 mutations determine a well-known lack of efficacy for the traditional anti-HER2 drugs [32]. Indeed, variants in the structure of HER2 proteins may hinder the correct function of these agents. One example is the expression of specific HER2 splicing variants, which compromise the successful binding of monoclonal antibodies such as trastuzumab. Specifically, when the extracellular domain of HER2 is cleaved by the metalloproteinase ADAM10, the result is the 95HER2 isoform, which is unable to bind antibodies, but is sensitive to TKIs such as lapatinib [33]. Given the absence of standardized methods to detect HER2 variants, their presence is not routinely assessed [31]. Another proposed mechanism of impaired binding for monoclonal antibodies is the masking of the HER2 binding site. For example, membrane-associated mucin 4 (MUC4), which could be expressed by tumour cells or other cells in the tumour microenvironment (TME), hides the trastuzumab binding site on the extracellular domain IV of HER2 [31,34].

### 3.2. HER2 Mutations

Activating mutations in the *ERBB2* gene are detected in 2–3% of primary BCs (regardless of HER2 amplification) and in 3–5% of mBC; they are more frequent in lobular cancer and are associated with worse prognosis [35,36,37,38]. The majority of *HER2* mutations are located in the tyrosine kinase and in extracellular domains [39]. In clinical practice, these variants cannot be detected by standard assays (IHC and FISH) and may only be detected through genome sequencing. Pre-clinical and clinical data support the hypothesis that these mutations have a role in the resistance to anti-HER2 treatment and to endocrine therapy [36,40].

In particular, regarding HER2-positive BC, variants of the intracellular domain confer specific resistance to anti-HER2 TKIs, determining their inability to block HER2 function. L755S is the most common mutation and represents a mechanism of resistance to lapatinib [41], but it could be overcame by neratinib [27,41]. Mutations L755S, V777L, D769Y and K753E are responsible for resistance to trastuzumab [36].

In HR-positive HER2-negative BC, somatic *HER2* mutations may lead to resistance to endocrine therapy [39,42], potentially conferring sensitivity to anti-HER2 TKIs [43]. The SUMMIT trial (NCT01953926) showed that the combination of neratinib plus fulvestrant is clinically active in heavily pre-treated *HER2*-mutant HR-positive mBC patients (including those pre-treated with fulvestrant and CDK4/6 inhibitors): ORR was 30%, clinical benefit rate was 47% and mPFS was 5.4 months [44,45,46]. Since genomic analyses suggested that resistance to neratinib may occur via mutant allele amplification or secondary *HER2* mutations, the trial was amended to explore dual HER2 targeting with the combination of neratinib, trastuzumab plus in this subgroup. This strategy showed a clinical benefit rate of 47% and a mPFS of 8.2 months [46,47].

### 3.3. Altered Intracellular Signalling

It is reasonable that resistance to anti-HER2 treatment derives from the constitutive activation of downstream signalling pathways of HER2. In this way, HER2-amplified cells become HER2-indipendent for their proliferation and survival, and the inhibition of the signalling cascade given by anti-HER2 treatment is bypassed. One example is represented by activating mutations in the alfa catalytic subunit of PI3K (PI3KCA), found in up to 20% of HER2-positive BCs [48,49]. This gene is known to be involved in PI3K/AKT/mTOR cascade, activated by HER2 overexpression, and its upregulation may lead to increased cell proliferation and tumour progression [50,51]. Recent evidence demonstrates that enrichment of MAP kinase pathway mutations can promote a switch in pathway dependence from PI3K/AKT to MEK/ERK, leading to resistance to anti-HER2 therapies [52]. In addition, parallel pathways may be boosted; for instance, the hyperactivity of ER enhances the PI3K/AKT/mTOR pathway and determines acquired resistance to anti-HER2 targeted treatment [53]. Aberrant activation of tyrosine kinase SRC and enhancement of the Cyclin D1-CDK 4/6 (cyclin dependent kinase) axis are other mechanisms of resistance described for both trastuzumab and TKIs [54]. Moreover, different RTKs can heterodimerize with HER2, especially EGFR/HER1 and HER3; the overexpression of these partners promotes the formation of HER2/EGFR and HER2/HER3 heterodimers, escaping the inhibition of HER2 homodimerization given by monoclonal antibodies. The formation of heterodimers is stimulated by neuroregulin-1 (NRG1), which has been associated with resistance to T-DM1 and to TKIs [31]. A crosstalk between the ER and the HER2 pathway has been described; indeed, approximately 50% of HER2-positive BCs overexpress ER and the crosstalk affect response to therapy and outcome. Finally, the overexpression of other proteins involved in the balance between cell death and survival could compensate the loss of HER2 function [31].

### 3.4. Other Mechanisms of Resistance

Anti-HER2 therapy is subject to multidrug resistance mechanisms, specifically caused by P-glycoprotein, ABCG2 (more commonly known as BCRP (Breast Cancer Resistance Protein)), and multidrug resistance protein (MRP). The presence of drug efflux pumps on the tumour cell wall also reduces the intracellular cytotoxic action of the warhead of ADCs (for example, DXd and DM). Moreover, the heterogeneity in the expression of HER2 could impair the effectiveness of anti-HER2 drugs [31,55].

Among trastuzumab’s mechanisms of action, one of the most important is its ability to trigger antibody-dependent cytotoxicity (ADCC), which occurs when trastuzumab is bound to breast cancer cells and its Fcγ region is recognized by immune cells expressing FcγRs. Several factors, such as polymorphisms in the Fcy receptor, may impair this binding and the immune activity of trastuzumab. The quantity of immune cells that are present in TME could also modulate the ADCC activity [56]; thus, combinations of anti-HER2 treatment and immunotherapy are being evaluated in several trials, so far with little success [12]. In this regard, the composition of the TME, which includes the surrounding non-cancerous cells and the extracellular matrix, is crucial for the development of the resistance to anti-HER therapies [57]. In order to maximize ADCC activity, a new agent was developed and recently approved by the FDA: margetuximab, a human/mouse chimeric IgG1 anit-HER2 monoclonal antibody. This drug has a trastuzumab backbone, with an engineered Fc-domain, in which the substitution of five amino acids boosts binding to FcyRIIIA (CD16A, a low affinity stimulatory receptor of macrophages and natural killer cells), and reduces the binding to the inhibitory Fc receptor FcYRIIB (CD32B) [58]. 

In order to overcome some of the resistances described above, a combination of anti-HER2 targeted and non-targeted therapies are currently used in clinical practice (for example, pertuzumab, trastuzumab and chemotherapy). Albeit the reasons for the resistance to anti-HER2 antibody combinations are currently unknown, some hypotheses indicate that a constitutive activation of downstream signalling pathways may play a major role [31].

## 4. HER Heterogeneity, HER2 Spatial Distribution and Anti-HER2 Therapy Resistance

As already mentioned, one of the possible mechanisms of resistance to anti-HER2 therapies is the intratumour heterogeneity in HER2 spatial distribution, that has been deeply investigated in BC [17]. In this paragraph, main definitions of HER2 heterogeneity and HER2 spatial distribution are outlined; a brief summary of the 2018 ASCO/CAP guidelines for HER2 evaluation is provided for the reader in Table 1.

HER2 genetic heterogeneity is defined as subclonal diversity within the tumour with an overall reported incidence from 5% to 30% [59] and in 1–34% of HER2-positive BCs [5]. Diagnostic guidelines have proposed different definitions for this concept [59,60,61]. It is defined by ASCO/CAP (2013) guidelines as the presence of ≥10% to <50% tumour cells with a ratio ≥ 2.0 when using dual probes or ≥6 HER2 signals/cell when using single probes, selecting 2–4 representative invasive tumour areas [16]. Vance et al. define a tumour as heterogeneous if there are more than 5% but less than 50% of infiltrating tumour cells with an ISH ratio higher than 2.2 [59]. Regarding the HER2 spatial distribution, three distinct patterns of distribution of cells with heterogeneous HER2 expression have been described: “clustered type”, “mosaic type” and “scattered type” (Figure 2). While in the first type two different tumour clones (one with HER2 amplification and the other with normal HER2 status) could be identified, the second type displays a diffuse intermingling of cells with different HER2 expression. The latter type is characterized by isolated HER-positive cells in a HER2-negative field [5].

To overcome this issue, and to categorise these tumours in the two main groups used in clinical practice (positive vs. negative), current guidelines consider BC samples as HER2-positive by IHC testing if there is an aggregate population of amplified cells composing > 10% of the total tumour cell population [4]. The recently introduced concept of HER2-low expression level has not been yet defined by ASCO/CAP guidelines, although these patients have been shown to benefit from ADCs [62]. Probably, a possible low intensity of HER2-expression should be taken into account while handling IHC assay results to ascertain the heterogeneity. 

Discordance of HER2 expression between primary and residual tumour has been reported after the neoadjuvant therapy and has been associated with a lack of pathologic complete response (pCR) [63]. In BC, therapy administration may lead to partial loss of selected cell populations, in particular HER2-targeted treatment may result in death of HER2-positive cell populations and survival of HER2-negative cells, resulting in HER2-heterogeneity [50]. Interestingly, the study of Filho et al. assessed HER2 heterogeneity in different areas of pre-treated HER2-positive early BC biopsies followed by T-DM1 administration, where 10% of samples were classified as those with heterogenous HER2 expression, none of which achieved a pCR compared to non-heterogenous BCs, where pCR was achieved in 55% of the cases [15]. The so-called “Subtype switch” after neoadjuvant treatment has been frequently observed and reported in literature with the highest frequency of HER2-enriched-to-luminal-A, and this effect has been found to be reversible after anti-HER2 therapy discontinuation [64].

Finally, when performing the IHC HER2-testing assay, one should be aware of possible errors, that may result in staining heterogeneity, due to various pre-analytical issues, such as fixation time, antibody clones selection and antigen retrieval systems [65]. To overcome the above-mentioned issues, a rigorous control and laboratory standardization is mandatory [65].

## 5. HER2 Heterogeneity and Response to Different Anti-HER2 Treatments

The pathological feature of HER2 heterogeneity has deep clinical implications, although its definition is still under debate [5]. Indeed, tumours with HER2 heterogeneity have shown to be less sensitive to anti-HER2 targeted treatments, both in metastatic and in early settings. For example, a shorter time to progression and lower OS has been reported during treatment with trastuzumab [66]. Novel drugs, such as ADCs, have raised intriguing hints [67]. Table 2 recapitulates these drugs and their mechanism of action.

In the MARIANNE clinical trial, T-DM1 showed non-inferior, but not superior, efficacy and better tolerability compared to taxane plus trastuzumab for first-line treatment of HER2-positive advanced BC [68]. A post-hoc exploratory analysis suggested that homogeneous HER2 IHC staining was associated with numerically longer median PFS than focal/heterogeneous staining in the HER2-positive population (IHC 2+ or 3+) [69].

In early-stage BC, HER2 heterogeneity is an independent factor predicting incomplete response to anti-HER2 neoadjuvant chemotherapy [70]. In the KRISTINE trial, neoadjuvant T-DM1 plus pertuzumab led to a lower pCR rate (44.4% vs. 55.7%; *p* = 0.016), compared to docetaxel, carboplatin, trastuzumab plus pertuzumab in HER2-positive stage II-III BC. Authors highlighted that BC samples from the 15 patients who experienced locoregional progression had lower HER2 expression and higher HER2 heterogeneity as compared to those from other patients in the T-DM1 arm [71]. The already mentioned study by Filho et al. was a phase II clinical trial, specifically designed to prospectively assess the impact of HER2 heterogeneity on response to targeted therapy. Authors defined HER2 heterogeneity as an area of HER2 amplification in more than 5% but less than 50% of tumour cells, or a HER2 negative area by ISH. One hundred sixty-four patients with centrally confirmed HER2-positive early-stage BC were treated with neoadjuvant T-DM1 plus pertuzumab. HER2 heterogeneity was detected in 10% of the cases; pCR was 55% in the non-heterogeneous subgroup and 0% in the heterogeneous group (*p* < 0.0001). Single cell *ERBB2* FISH identified the fraction of HER2 non amplified cells as the driver of resistance [15]. In another study, 37 HER2+ BCs were analysed first with combined immunofluorescence and ISH followed by a validated computational approach. Samples were analysed before and after neoadjuvant HER2-based treatment, in order to study tumour evolution as well. This study confirmed that heterogeneous tumours were associated with significantly shorter DFS and fewer long term survivors [18]. These data suggest that in the absence of systemic chemotherapy, the bystander killing effect (i.e., the unintentional payload diffusion from antigen-positive tumour cells to adjacent antigen-negative tumour cells) of T-DM1 may not be sufficient to eradicate heterogeneous HER2 BC. Novel ADCs, such as T-DXd, have been specifically designed to enhance the possibility to have a bystander activity: preclinical evidence highlights the cruciality of a cleavable linker that could release the payload from the antibody moiety and of a hydrophobic payload that could diffuse through the cell membrane towards neighbouring cells. In this way, the cytotoxic compound could be potentially delivered also to antigen-negative cells, providing higher chances of efficacy on heterogeneous tumours [24,72] (Table 3). 

However, another layer of complexity should be added. Indeed, in all the cases reported above, BC could be heterogeneous or not heterogeneous, but remains still HER2-positive per definition (IHC 3+ and/or FISH amplified). Recently, the evidence that a bystander killing effect could play a role also in HER2-low and HER2-null (IHC 0) tumours set the biological rationale for ambitious translational and clinical studies. One example is the phase 3 DESTINY-Breast04 trial, in which T-DXd efficacy was evaluated in 557 heavily pre-treated patients with HER2-low expressing advanced BC (494 ER-positive, 63 ER-negative) [74]. In the ER-positive cohort, the mPFS was 10.1 vs. 5.4 months (hazard ratio = 0.51, *p* ≤ 0.001), and the OS was 23.9 vs. 17.5 months, favouring T-DXd compared with treatment of physician choice (hazard ratio 0.64, *p* = 0.003). In the intention-to-treat (ITT) population, median PFS was 9.9 vs. 5.1 months, (hazard ratio = 0.50, *p* ≤ 0.001), and OS was 23.4 vs. 16.8 months (hazard ratio 0.64, *p* = 0.001). According to these results, T-DXd has been approved by the FDA and the EMA for the treatment of HER2-low BC patients. 

Furthermore, in the phase 2 DAISY clinical trial (NCT04132960) 179 patients with heavily pre-treated advanced BC were treated with T-DXd in order to assess the activity of the drug, its mechanism of action, and to identify biomarkers associated with drug response or drug efficacy. The study design provided three arms: Cohort 1, HER2 IHC 3+ (*n* = 68); Cohort 2, HER2-low (*n* = 73); Cohort 3, HER-null IHC 0 (*n* = 38) [73]. Biopsy of metastatic sites was performed at baseline, during treatment and at tumour progression. Primary endpoint was the best overall response in each cohort. At a median follow-up of 15 months, the overall response in the HER2-positive cohort was 70%, 37.5% and 29.7% in HER2-positive, HER2-low and HER2-null cohort, respectively. mPFS was 11.1, 6.7 months (6.9 in HR-positive and 3.5 in HR-negative) and 4.2 months (4.5 in HR-positive and 2.1 in HR-negative), for each cohort. During the European Society of Medical Oncology (ESMO) Breast Cancer Congress 2022, a translational analysis of the DAISY trial was disclosed, investigating T-DXd efficacy according to HER2 expression status [80]. These molecular analyses showed that cancer cells expressing low levels of HER2, but not HER2 IHC 0 cells, can internalize T-DXd (*p* < 0.053). By applying artificial intelligence to digital pathology, the predictive value of HER2 spatial distribution was investigated using weakly supervised and clustering algorithms on HER2-positive whole slide imaging (WSI) at baseline (*n* = 61). Clustering algorithms in the HER2-positive cohort identified a cluster associated with a lower best overall response (BOR) (*p* < 0.007), demonstrating that a high percentage of HER2 IHC 0 cells and their spatial distribution were associated with no response to treatment (*p* = 0.0008). At progression, a decreased HER2 expression was observed in 13/25 (52%) patients (*p* < 0.07) [81]. Although these data are intriguing, results from the full publication are awaited. 

## 6. Further Directions

The study of the heterogeneity and spatial distribution of HER2 has brought some important results but has also raised some intriguing clinical hints for the treatment of HER2-positive and HER2-low BC. In particular, the advent of ADCs has disclosed the possibility to overcome HER2 intratumoural heterogeneity. Further studies on the definition of HER2 heterogeneity are mandatory to improve patients’ outcomes. Three main issues could be identified in order to address further research: (1) improving the assessment of HER2 heterogeneity; (2) dissecting the complexity of the interaction between HER2-positive cells and other cell types in the TME; (3) identifying new drugs capable of overcoming resistance. 

Firstly, nowadays HER2 assessment is based on a qualitative IHC evaluation. Therefore, the study of quantitative assays to evaluate HER2 is currently under investigation. For instance, one study provides the use of The HERmark™ Breast Cancer Assay (HERmark, Monogram Biosciences, South San Francisco, CA, USA), that is a quantitative method with a dual-antibody, proximity-based approach. This evaluation provides accurate quantification of the HER2 protein in tissue samples using a dual-antibody, proximity-based immunoassay approach, the VeraTag™ technology (Monogram Biosciences), to make precise and quantitative measurement of total HER2 protein expression with greater sensitivity and specificity than IHC [82]. Moreover, other possibilities to assess HER2 are looming on the horizon. For instance, intriguing clinical trials (NCT02095210) are evaluating the uptake distribution of the HER2-binding radiolabelled agent [68Ga] ABY-025 by PET imaging in patients harbouring a biopsy-identified HER2 expression.

Secondly, some intriguing analysis using spatial transcriptomic technologies have already been performed to study the spatial-gene expression profiles of HER2-positive tumours [83]. Moreover, spatial proteomic characterization of HER2-positive BCs through neoadjuvant chemotherapy was demonstrated to be predictive of response in 57 HER2-positive BCs from the neoadjuvant TRIO-US B07 clinical trial. In this trial, samples were collected pre-treatment, after 14–21 days of HER2-therapy and at surgery. The proteomic changes after one cycle of therapy could predict pCR, outperforming transcriptomic changes [84].

Single cell analysis has also been performed to identify HER2-low patients by capturing the spatial distribution of HER2 expression across the entire tumour from a subset of samples from ISPY1 and 2 trials. In this way, researchers could identify 84 patients affected by triple negative breast cancer (TNBC) expressing HER2-low who could potentially benefit from anti-HER2 drugs [85]. Among them, 17 patients expressed HER2 at low levels and a previously published HER2-responsive gene expression signature [85].

Finally, new pharmacological approaches have been proposed. In particular, mathematical models could be used in order to hypothesize which treatment combinations and schedules could be more effective in this patient population [17]. Possible approaches exploit the combination of HER2-targeting and non-targeting agents and the use of pharmacological compounds capable of reducing intratumoural heterogeneity and phenotypic plasticity, such as histone deacetylase (HDAC), histone demethylase, and bromodomain inhibitors [17]. Innovative platforms, such as bispecific antibodies, may combine proprieties of different drugs in order to overcome resistance. An example is zanidatamab, a humanised, bispecific monoclonal antibody directed against two non-overlapping domains of HER2 (i.e., the trastuzumab binding domain and the pertuzumab binding domain) that is currently under investigation for early stage and metastatic HER2-positive BC (NCT05035836, NCT04224272, NCT02892123). Zanidatamab zovodotin (zanidatamab-based antibody conjugated with zovodotin, an auristatin cytotoxic agent) is also being tested in a phase I trial (NCT03821233) and may represent a possible way to overcome the resistance determined by the heterogeneous expression of HER2. Another attempt supports adaptive therapy, in which the competition of drug-resistant and drug-sensitive cells is limited by alternating therapy and no treatment [17].

## 7. Conclusions

The discovery of the HER2 protein and anti-HER2 targeted treatments has completely changed the prognosis of a subgroup of patients affected by BC. Since the introduction of trastuzumab in clinical practice, the treatment armamentarium has been expanded with the development of new pharmacological compounds such as the novel ADCs, resulting in progressive and important improvements in clinical outcomes for patients. The efforts of the scientific community should be addressed to a sharper dissection of this entity and to fully understand the implications of the heterogeneity in HER2 expression and its spatial distribution. In this context, new horizons will be defined by novel methods to dissect BC heterogeneity and innovative drugs targeted to overcome these mechanisms of resistance, such as modulators of histone proteins or bispecific antibody–drug conjugates. In this way, the best treatment choice could possibly be achieved, potentially de-escalating therapies in those patients with HER2-homogeneous expressing tumours and escalating treatment in HER2 heterogeneous ones. 

## Figures and Tables

**Figure 1 cancers-15-01385-f001:**
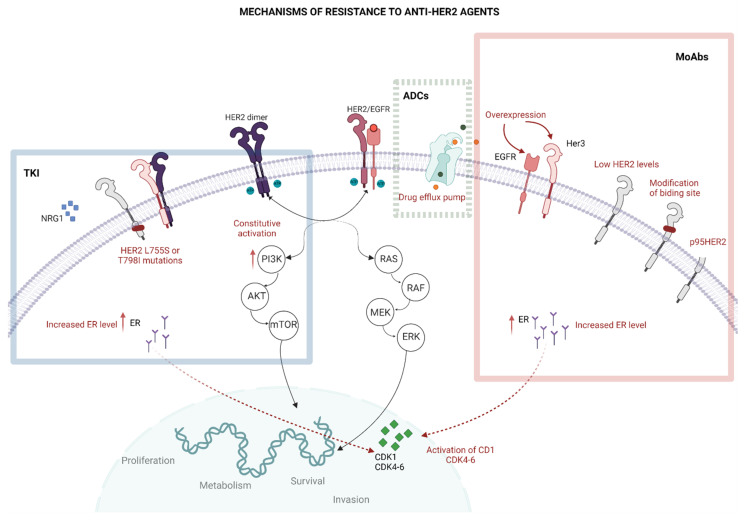
Mechanism of resistance to anti-HER2 targeted therapy. (1) Resistance to monoclonal antibodies (right, red box). Numerous mechanisms of resistance have been described. First, low levels of HER2 expression determine few receptors to bind for Ab, thus lowering the effectiveness of Ab. Changes of the extracellular binding domain could generate resistance as well: the most studied variant is p95HER2, an isoform deriving from the cleavage of the extracellular domain by ADAM10 that causes the impossibility of HER2 binding for trastuzumab. Another mechanism of resistance is the constitutive activation of the PI3K pathway (not shown in the box), depending on the alteration of the alpha catalytic subunit of PI3K. Different RTKs can heterodimerize with HER2, especially EGFR and HER3; their overexpression promotes the formation of HER2–EGFR and HER2–HER3 heterodimers, activating the pathway. The increase in the ER expression, as well as the alterations of CD1 or CDK4-6, stimulates cell growth and survival, reducing the inhibitory effect of monoclonal antibodies. (2) Resistance to ADCs (centre, green box): besides the mechanisms of resistance shared with other classes of drugs, some escapes are peculiar for ADCs. For example, the presence of drug efflux pumps on the cell wall reduces the intracellular cytotoxic action of the warhead of ADCs (for example, Dxd and DM). NRG-1 overexpression stimulates the dimerization of EGFR–HER3 and the activation of PI3K pathways in tumours treated with T-DM1 (not shown in the figure). (3) Resistance to TKIs (left, blue box): low HER-2 expression, activation of CD1 and CDK4-6, increased ER, constitutive activation of PI3K pathway and the overexpression of NRG1 are mechanisms of resistance shared with other drug classes. Mutations of the binding site in the intracellular domain confer specific resistance, determining the inability of TKIs to block HER2 function. The most frequent mutations are L755S and T798I, that cause resistance to lapatinib. Abbreviations: ADCs: antibody–drug conjugates; EGFR: epidermal growth factor receptor; HER2: human epidermal growth factor receptor 2; ER: estrogen receptor; NRG1: neuro regulin 1; CDK: cyclin dependent kinase. Created with BioRender.com (www.biorender.com, accessed on 23 December 2022).

**Figure 2 cancers-15-01385-f002:**
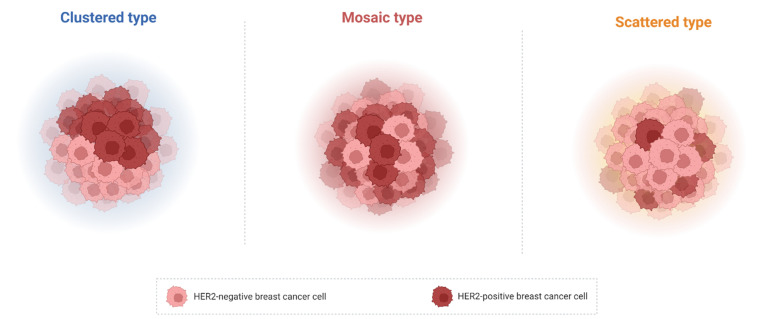
Patterns of distribution of cells with heterogeneous HER2 status. Clustered type (**left** panel): two different tumour clones, one with HER2 amplification and the other with normal HER2 status, could be identified. Mosaic type (**central** panel): diffuse intermingling of cells with different HER2 expression. Scattered type (**right** panel): isolated HER-positive cells in a HER2-negative field. Abbreviations: HER2: human epidermal growth factor receptor 2. Created with BioRender.com (www.biorender.com, accessed on 4 February 2023).

**Table 1 cancers-15-01385-t001:** Algorithm for evaluation of HER2 expression according to the 2018 ASCO/CAP guidelines.

HER2 Status	Score IHC (Definition)	ISH (Definition) °^
**HER2-positive**	3+ (Circumferential membrane staining, complete, intense, in > 10% of TCs)	Amplified (Average HER2 copy number ≥ 6.0 signals/cell)
	2+ * (Weak/moderate complete membrane staining in > 10% of TCs)	
**HER2-negative**	2+ ^#^ (Weak/moderate complete membrane staining in > 10% of TCs)	Not Amplified (Average HER2 copy number < 4.0 signals/cell)
	1+ (Incomplete membrane staining, faint/barely perceptible in > 10% of TCs)	
	0 (No staining or incomplete membrane staining, faint/barely perceptible in ≤10% of TCs)	

Abbreviations: TC: tumour cells; HER2: human epidermal growth factor receptor 2; *: if ISH positive; #: if ISH negative; °: Note: in clinical practice, the evaluation of the status of HER2 by ISH is performed only when IHC score is 2+; ^: if ISH result is between 4 and 6, breast cancer is classified as HER2-negative.

**Table 2 cancers-15-01385-t002:** Antibody–drug conjugates targeting HER2 currently available for the treatment of breast cancer.

Drug	Antibody	Linker	Payload	Bystander Effect	Indication
**Trastuzumab emtansine**	Trastuzumab	Non-cleavable	DM1 (microtubule inhibitor)	No	HER2-positive BC
**Trastuzumab deruxtecan**	Trastuzumab	Cleavable	DXd (topoisomerase-1 inhibitor)	Yes	HER2-positive BC HER2-low BC

Abbreviations: DXd: deruxtecan; DM1: emtansine; HER2: human epidermal growth factor receptor 2.

**Table 3 cancers-15-01385-t003:** Summary of trials investigating agents in heterogeneous HER2 expression in breast cancer.

HER2 Expression	Treatment	Trial	Phase	Setting	Primary Endpoint	Ref.
**HER2-negative** **(HER2 IHC: 0)**	T-DXd	DAISY (NCT04132960)	2	Advanced	BOR 30.6% (95%CI: 22.7–45.4)	[73]
**HER2-Low** **(HER2 IHC 1+ OR HER2 IHC 2+ ISH non ampl)**	T-DXd	DAISY (NCT04132960)	2	Advanced	BOR 33.3% (95%CI: 56.7–79.8)	[73]
	T-DXd vs. TPC	DESTINY-Breast06 (NCT04494425)	3	Advanced	Ongoing	//
	T-DXd vs. TPC	DESTINY- Breast04 (NCT03734029)	3	Advanced (>1–2 line)	mPFS * 9.9mo vs. 4.9mo HR 0.50 (95% CI: 0.40–0.63; *p* < 0.0001)	[74]
**HER2-positive** **(HER2 IHC 2+ and ISH ampl OR HER2 IHC: 3+)**	T-DM1 vs. lapatinib + capecitabine	EMILIA (NCT00829166)	3	Advanced (2nd line)	mPFS 9.6 mo vs. 6.4 mo; HR: 0.65 (95% CI, 0.55–0.77, *p* < 0.001)	[75]
	T-DM1 vs. TTZ + taxane	MARIANNE (NCT01120184)	3	Advanced (1st line)	mPFS 14.1 mo vs. 13.7 mo (non-inferiority) HR: 0.91 (97.5% CI, 0.73–1.13)	[68]
	T-DM1 vs. TTZ	KATHERINE (NCT01772472)	3	Post-neoadjuvant	13y-iDFS 88.3% vs. 77%; HR: 0.50 (95% CI, 0.39–0.64; *p* < 0.001)	[21,76]
	T-Dxd	DAISY (NCT04132960)	2	advanced	BOR 69.1% (95% CI: 39.1–54.2)	[73]
	Combinations of T-DXd and other agents	DESTINY-Breast07 (NCT04538742)	1b-2	Advanced (1st line)	ongoing	//
	T-DXd vs. T-DM1	DESTINY-Breast03	3	Advanced (2nd line)	mPFS 28.8 vs. 6.8 mo; HR: 0.33 (95% CI, *p* = <0.0001)	[24,25]
	T-DXd	DESTINY-Breast01 (NCT03248492)	2	Advanced (≥3 line)	ORR 60.9% (95% CI, 53.4–68)	[77]
	Trastuzumab Duocarmazine	TULIP (NCT03262935)	3	Advanced (≥2 line)	mPFS 7 mo (95% CI, 5.4–7.2) vs. 4.9 mo TPC (4–5.5); HR: 0.64 (95% CI, 0.49–0.84, *p* = 0.002)	[78]
	ARX788	ACE-Breast01 and ACE-Pantumour01 (NCT03255070)	1	Advanced	ORR 74% (14/19) ACE-Breast-01 67% (2/3) ACE-Pan tumour-01	[79]
	T-DM1 + pertuzumab	NA	2	neoadjuvant	pCR 55% in the non heterogeneous subgroup and 0% in the heterogeneous group (*p* < 0.0001) ^#^	[15]

Abbreviations: T-DXd: trastuzumab deruxtecan; T-DM1: trastuzumab emtansine; HER2: human epidermal growth factor receptor 2; ISH: in situ hybridization; TPC: treatment of physician choice; mo: months, TTZ: trastuzumab; iDFS invasive disease-free survival; ORR: objective response rate, IHC: immunohistochemistry, ampl: amplifed, HR: hazard ratio, BOR: Best Overall Response, PFS: Progression Free Survival; OS: overall survival; HR: hazard ratio; * among all patients; #: adjusted per hormone receptor status; Ref: reference.

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
