# Peer review of "Unlocking the Resistance to Anti-HER2 Treatments in Breast Cancer: The Issue of HER2 Spatial Distribution"

_cancers, 2023, doi:10.3390/cancers15051385_

Round 1

Reviewer 1 Report

  1. The authors write that new pharmacological agents, belonging to the group of antibody-drug conjugates, may overcome resistance to anti-HER2 therapy. It is desirable to add a table with such a drugs and their mechanism of action. In a line 335 authors write about it : “3) identifying new drugs capable to overcome  the resistance”. And only 1 sentence is about possible drugs (histone deacetylase) etc
  2. The authors don’t write about multidrug resistance, particularly, caused by P-glycoprotein, BCRP, MRP, that can be also a reason of the resistance, as well as mutations in HER2 receptor. The polymorphism of the HER2 gene and other pharmacogenetic aspects of the resistance should be mentioned in a more detailed manner. What are the recommended procedures before the administration of the anti-HER2 therapy?
  3. In this review the main aspect of the resistance to anti-HER2 treatments in breast cancer is belong to HER2 spatial distribution. But not much attention is paid to the perspective and quick methods of it identification. Also the relationship between HER2 heterogeneity and endocrine resistance is not discussed.
  4. There is mistake “ 3.1. mpaired binding to HER2”
  5. Is the ABB “MoAbs” frequently used term?
  6. “Conclusions” and “Abstract” do not contain concrete ways to overcome resistance, concrete strategies or drugs. It`s desirable
  7. For the “review” it is too few links, only 65
  8. Section 4: it is desirable to illustrate the main idea of the chapter.
  9. line 405: “omogeneous expressing tumours and escalating treatment in HER2 heterogeneous ones”.

Reviewer 2 Report

In this narrative review Giugliano et. al. recapitulate the available evidence about resistance to anti-HER2 compounds in breast cancer, with a specific focus on HER2 spatial heterogeneity. This represents a hot topic in the field of breast cancer. Despite a plethora of reviews about HER2 heterogeneity have been published  in the last 3 years, the present one address this issue under the specific standpoint of treatments' resistance.

The manuscript is well written and adequately structured. The topic is exposed clearly and concisely, with proper literature citations.

However, I recommend some minor revisions before publications:

1. Page 4, line 167: the "i" of "impaired" is missing;

2. Page 5, line 205: a reference should be added at the end of the sentence;

3. Page 5, line 212: the word "polymorphisms" should be cut out;

4. Page 5, line 215: missing full stop after ref n° 25;

5. I strongly suggest to add a second figure to display the different pathways of HER2 heterogeneous expression (“clustered type”, “mosaic type” and “scattered type”);

6. Table 1. HER2 IHC 1+ should be added in the definition of HER2 low status. HER2 IHC 3+ and HER2 IHC 2+; ISH ampl should be also added to define HER2-positive, in order to maintain consistency. 

7. I suggest to revise the text punctuation. For example, references should be placed before full stops and not after them.

Reviewer 3 Report

This review summarizes current evidence on the role of HER2 heterogeneity and spatial distribution to the current available treatment choices of anti-HER2 treatments. The review is important and significant although the resistance to anti-HER2 treatment has been widely evaluated. However, it will be important to include the novelty/significance of this review comparing to the previous relevant reviews such as (doi.org/10.3390/cancers14163996; 10.18632/oncotarget.7043) in the introduction. 

Reviewer 4 Report

The authors have written a fine review on the mechanism of resistance to anti-HER2 targeted therapy in HER2 positive breast cancers. The authors also highlighted the effect of spatial heterogeneity of HER2 expression on treatment response and assessment of HER2 status.

Authors discuss novel treatment strategies that take into account the heterogeneity in HER2 expression and pay special attention to their current clinical development and summarize relevant clinical trials. The only minor issues are spelling errors. For example, in Figure1 legend there are a few mistakes:  .the most studied variant…, domain byby ADAM10... Also 3.1 subtitle has a missing letter. The authors should check the paper again for minor language mistakes.

I recommend acceptance of this paper for publication in Cancers.

Reviewer 5 Report

In this article, Giugliano et al, provide a comprehensive review on the topic of HER2 heterogeneity in breast cancer. HER2 heterogeneity presents a significant clinical challenge with a profound impact on response to and choice of HER2-targeted therapies and overall clinical outcomes.

The authors first give an overview of the current management of HER2-positive breast cancer and the mechanisms of resistance to HER2-targeted therapies. Then, they delve into the current understanding of HER2 heterogeneity, including its definition, relationship to anti-HER2 therapy resistance, and response to different HER2-targeted therapies. Finally, they offer a comprehensive overview of future diagnosis and treatment of HER2 heterogeneous tumors.

Overall, this well-written review provides an up-to-date summary of the knowledge on HER2 heterogeneity in breast cancer. The citations are comprehensive and accurate. I only have a few minor suggestions to the authors.

1.      In chapter #4, it will be better to attach an updated table of the 2018 ASCO/CAP guidelines for HER2 evaluation by ISH to help readers better understand the current criteria for diagnosing HER2 heterogeneity. 

2.      The authors need to make a few changes to the titles of chapter #4 and #5 to better reflect the content, including: Chapter 4 – HER2 heterogeneity and anti-HER2 therapy resistance, and Chapter 5 – HER2 heterogeneity and response to different anti-HER2 treatment. 

Round 2

Reviewer 1 Report

The article have been significantly improved.

It is recommended to clarify or describe the position and function of NRG1 on Fig.1

Round 3

Reviewer 1 Report

Dear Authors,

As it seems to me, the role of NRG 1 in Fig.1 is not very clear to the readers.

As follows from the figure, NRG 1 is transported by the drug efflux pump.

As a suggestion, not a comment: is it possible to separate NRG 1 and drug efflux pump and draw an arrow from NRG 1.
